# Retinal BMI1 Expression Preserves Photoreceptors in Sodium-Iodate-Induced Oxidative Stress Models

**DOI:** 10.3390/ijms26125907

**Published:** 2025-06-19

**Authors:** Zhongyang Lu, Shufeng Liu, Maria G. Morales, Andy Whitlock, Ram Ramkumar, Hema L. Ramkumar

**Affiliations:** Oculogenex Inc., 2250 W Whittier Blvd., Suite 300, La Habra, CA 90631, USA; sliu@oculogenex.com (S.L.); mgmorales@gmail.com (M.G.M.); awhitlock@oculogenex.com (A.W.);

**Keywords:** AAV gene therapy, BMI1, retina, retinal pigment epithelium, sodium iodate, age-related macular degeneration

## Abstract

Dry age-related macular degeneration (AMD) is a leading cause of vision loss in individuals over 50, yet no approved therapies exist for early or intermediate stages of the disease. Oxidative stress is a central driver of retinal degeneration in AMD, and sodium iodate (NaIO_3_)-induced injury serves as a well-characterized model of oxidative damage to the retinal pigment epithelium (RPE) and photoreceptors. BMI1, a poly-comb group protein involved in DNA repair, mitochondrial function, and cellular renewal, has emerged as a promising therapeutic target for retinal neuroprotection. We evaluated the efficacy of AAV-mediated BMI1 gene delivery in murine models using two administration routes: subretinal (SR) and suprachoroidal (SC). AAV5.BMI1 (1 × 10^9^ vg/eye) was delivered SR in Balb/c mice and evaluated at 4 and 15 weeks post-injection. AAV8.BMI1 (5 × 10^9^ or 1 × 10^10^ vg/eye) was administered SC in C57BL/6 mice and assessed at 4 weeks. Control groups received BSS or AAV8.stuffer. Following NaIO_3_ exposure, retinal structure and function were analyzed by optical coherence tomography (OCT), electroretinography (ERG), histology, and molecular assays. SC delivery of AAV8.BMI1 achieved the highest levels of retinal BMI1 expression with no evidence of local or systemic toxicity. Treated eyes showed dose-dependent preservation of outer nuclear layer (ONL) thickness and significantly improved ERG responses indicating structural and functional protection. These findings support SC AAV.BMI1 gene therapy as a promising, minimally invasive, and translatable approach for early intervention in intermediate AMD.

## 1. Introduction

Age-related macular degeneration (AMD) is the leading cause of irreversible vision loss in individuals over 50 years of age. In the worldwide, AMD affects an estimated 196 million people (95% CI: 140–260 million) as of 2019, with projections rising to 288 million by 2040 (95% CI: 205–399) [1,2]. Early and intermediate stages of dry AMD are marked by progressive dysfunction and degeneration of the retinal pigment epithelium (RPE) and photoreceptors. Despite the high prevalence and chronic progression of the disease, there are currently no FDA-approved therapies for early or intermediate AMD, and no existing interventions durably alter the natural course of nonexudative disease.

Recent therapeutic failures in AMD underscore the urgent need for novel, mechanism-based interventions. The complement factor D inhibitor lampalizumab failed to demonstrate efficacy in slowing geographic atrophy progression in the phase III SPECTRI and CHROMA trials [3]. Similarly, development of RO7171009, an antibody targeting HTRA1, was discontinued due to the lack of a meaningful clinical benefit in patients with geographic atrophy [4]. These examples highlight the limitations of current approaches and reinforce the need for durable strategies that target core pathogenic mechanisms such as oxidative stress and mitochondrial dysfunction. In this context, BMI1 gene therapy offers a promising alternative with the potential to modify disease progression at earlier stages.

BMI1 encodes a poly-comb group protein involved in DNA repair, mitochondrial function, oxidative stress regulation, and stem cell self-renewal [5,6,7,8]. While BMI1 expression shows a moderate decline in aged and early AMD RPE, its expression may enhance retinal resilience by compensating for stress-induced damage independent of baseline levels [9,10,11,12]. BMI1 has demonstrated protective effects in multiple tissues by preserving cellular integrity, reducing oxidative stress, and promoting mitochondrial health [13,14,15,16,17]. In the retina, BMI1 is known to regulate homeostasis and injury response pathways in RPE, photoreceptors, and Müller glia [5,12]. It modulates key regulators of cellular senescence and apoptosis, including the p16INK4a/Rb and p19ARF/p53 pathways [18,19], and may suppress gliosis via its expression in Müller cells [20]. Together, these cell-specific roles support the therapeutic rationale for targeting BMI1 in degenerative retinal diseases.

Oxidative stress and mitochondrial dysfunction are well-established contributors to AMD pathogenesis and are the focus of emerging treatment strategies [21,22]. The recent FDA approval of photobiomodulation devices (e.g., Valeda^®^) that target mitochondrial health underscores the translational relevance of these pathways. Given its role in DNA damage response and redox balance, BMI1 represents a compelling therapeutic target for dry AMD [11,23].

The sodium iodate (NaIO_3_)-induced retinal degeneration model recapitulates key histological features of oxidative injury seen in AMD, including RPE atrophy, photoreceptor loss, and outer retinal thinning [21,24,25]. These changes parallel the pathology of advanced nonexudative AMD and make this model suitable for evaluating interventions that target oxidative injury [26,27,28].

In this study, we investigated the therapeutic potential of BMI1 gene therapy delivered via adeno-associated virus (AAV) vectors in murine models of NaIO_3_-induced oxidative retinal degeneration. We compared subretinal (SR; AAV5.BMI1) and suprachoroidal (SC; AAV8.BMI1) routes of administration to evaluate their efficacy and translational potential. Two mouse strains (Balb/c and C57BL/6) were used to assess therapeutic durability across early and late timepoints. Retinal structure was quantified using outer nuclear layer (ONL) thickness—a validated surrogate of photoreceptor survival and visual function. This study addresses a critical gap in the treatment of dry AMD and supports the development of BMI1-based gene therapy as a durable, disease-modifying intervention for intermediate disease stages.

## 2. Results

### 2.1. Sodium Iodate Induces Severe Structural and Functional Retinal Degeneration in Mice

To validate the sodium iodate (NaIO_3_) model of oxidative retinal degeneration, Balb/c and C57BL/6 mice were administered NaIO_3_ intravenously at doses of 40 or 50 mg/kg. Retinal structural and functional integrity were assessed 4 weeks post-injection using optical coherence tomography (OCT) and electroretinography (ERG).

OCT imaging revealed profound disruption of the outer retinal architecture in NaIO_3_-treated mice compared with balanced salt solution (BSS) controls. Control animals exhibited preserved retinal layers, including intact inner segment/outer segment (IS/OS) junctions, outer nuclear layer (ONL), and retinal pigment epithelium (RPE). In contrast, NaIO_3_-treated mice showed loss of IS/OS integrity, RPE degeneration, and marked ONL thinning (Figure 1A).

Quantitative analysis confirmed significant ONL thinning in NaIO_3_-exposed animals. In the nasal retina, ONL thickness was reduced from 66.3 ± 4.6 µm in controls to 33.4 ± 8.7 µm in treated eyes; in the temporal retina, thickness decreased from 64.3 ± 3.3 µm to 29.8 ± 10.9 µm (Figure 1B; *n* = 10 eyes per group, *** *p* < 0.001).

Functional deficits were assessed using ERG. NaIO_3_-treated mice demonstrated severely diminished retinal function, with extinguished scotopic a-waves, photopic b-waves, and c-waves, indicating widespread dysfunction of photoreceptors, bipolar cells, and RPE (Figure 1C). All ERG responses were abolished by the 4-week endpoint, reflecting the severity of oxidative injury. This model induces extensive structural and functional damage and thus may present challenges for detecting functional protection unless therapeutics are administered prophylactically or the injury is attenuated.

### 2.2. Durable, Dose-Dependent Increases in Retinal BMI1 Expression Following AAV-Mediated Delivery

To evaluate transgene expression and dosing durability, BMI1 mRNA and protein levels were assessed in murine eyes treated with either subretinal AAV5.BMI1 or suprachoroidal AAV8.BMI1. Both delivery strategies led to durable expression of BMI1 in the retina and RPE, with the suprachoroidal route yielding dose-dependent and consistent upregulation.

#### 2.2.1. AAV5.BMI1 Subretinal Delivery Increases Retinal BMI1 mRNA Expression over Time

Subretinal delivery of AAV5.BMI1 (1 × 10^9^ vg/eye) in Balb/c mice resulted in significant and sustained upregulation of Bmi1 mRNA in the RPE at 4, 8, and 14 weeks post-injection compared with BSS controls (Figure 2A). While BMI1 protein levels also showed an upward trend in AAV5.BMI1-treated retinal tissues, high inter-animal variability limited statistical significance in protein quantification. This variability may reflect challenges in achieving consistent subretinal bleb formation and vector distribution in small animal models.

#### 2.2.2. AAV8.BMI1 Suprachoroidal Delivery Achieves Robust, Dose-Dependent BMI1 Expression

To assess BMI1 expression following suprachoroidal injection, C57BL/6 mice received AAV8.BMI1 at either 5 × 10^9^ or 1 × 10^10^ vg/eye or control injections of BSS or AAV8.stuffer (lacking any regulatory or coding elements). Four weeks later, animals were euthanized, and retinal tissues were harvested for microdissection and RNA/protein extraction.

Protein levels were measured using a validated electrochemiluminescence (ECL) assay (Meso Scale Discovery). AAV8.BMI1 treatment resulted in a significant, dose-dependent increase in BMI1 protein expression in both retina and RPE compared with controls (Figure 2B). Similarly, qRT-PCR analysis confirmed a significant increase in BMI1 mRNA levels in both compartments following high-dose AAV8.BMI1 (1 × 10^10^ vg/eye) administration (Figure 2C).

These results demonstrate that suprachoroidal delivery of AAV8.BMI1 leads to robust, reproducible transgene expression in both retinal and RPE tissues and supports dose optimization for translational applications.

### 2.3. BMI1 Expression Preserves Photoreceptors and Retinal Structure

Both subretinal and suprachoroidal delivery of AAV.BMI1 preserved retinal integrity and prevented photoreceptor degeneration in NaIO_3_-induced retinal injury models. In Balb/c mice, AAV5.BMI1 treatment significantly increased rhodopsin expression in photoreceptors, as detected by immunofluorescence (Figure 3A). Histological analysis performed 30 days post-AAV5.BMI1 injection and 3 days after NaIO_3_ exposure (50 mg/kg, intraperitoneal) demonstrated preservation of outer nuclear layer (ONL) thickness (44.5 ± 1.05 µm), compared with 22.6 ± 1.13 µm in NaIO_3_-only treated animals and 66.7 ± 1.53 µm in untreated controls (Figure 3B). These results indicate structural preservation of the retina with AAV5.BMI1 gene therapy.

AAV8.BMI1-treated C57BL/6 mice also exhibited significant protection against NaIO_3_-induced retinal degeneration. Mice receiving suprachoroidal AAV8.BMI1 followed by NaIO_3_ (40 mg/kg, intravenous) displayed strong nuclear BMI1 staining within the ONL on immunohistochemistry, indicating transgene expression in preserved photoreceptors (Figure 3C). Histological sections revealed robust ONL preservation and outer segment integrity compared with controls.

To further quantify structural preservation, OCT imaging was performed 4 weeks post-NaIO_3_ exposure. Both AAV5.BMI1 and AAV8.BMI1 treatment groups showed significantly greater ONL thickness compared with saline or AAV8.stuffer controls (Figure 3D–F). Quantitative analysis of ONL thickness measured 1500 µm on either side of the optic nerve confirmed a significant increase in retinal thickness in AAV8.BMI1-treated mice receiving 1 × 10^10^ vg/eye (*p* < 0.01, *n* = 5 per group).

These data collectively demonstrate that BMI1 gene therapy preserves the retinal architecture and protects photoreceptors in models of oxidative retinal injury, with therapeutic effects evident across both delivery routes and mouse strains.

### 2.4. AAV8.BMI1 Treatment Preserves Retinal Function Following Sodium Iodate Exposure

To determine whether AAV8.BMI1 gene therapy preserves retinal function following oxidative injury, electroretinography (ERG) was performed in C57BL/6 mice treated with suprachoroidal (SC) AAV8.BMI1. Subretinal (SR) delivery is known to suppress ERG amplitudes in mice due to retinal detachment during injection; thus, functional analysis was not performed in AAV5.BMI1-treated animals.

ERG responses were recorded at baseline and four weeks following systemic NaIO_3_ administration (40 mg/kg, IV). Mice received SC injections of AAV8.BMI1 at either 5 × 10^9^ or 1 × 10^10^ vg/eye, with BSS and AAV8.stuffer groups serving as controls.

Compared with control groups, AAV8.BMI1-treated mice receiving the 1 × 10^10^ vg/eye dose exhibited significantly higher a-wave and b-wave amplitudes, indicating preserved photoreceptor and bipolar cell function (Figure 4). In contrast, the 5 × 10^9^ vg/eye dose did not yield significant improvements, suggesting that a threshold level of BMI1 expression is required for functional protection. These data demonstrate that high-dose AAV8.BMI1 gene therapy maintains retinal function under oxidative stress and supports dose optimization for translational applications.

## 3. Discussion

There are currently no FDA-approved, durable therapies that prevent progressive retinal degeneration due to chronic oxidative stress, which plays a central role in the pathogenesis of dry age-related macular degeneration (AMD). Numerous investigational therapies that showed promise in transgenic or acute injury models have failed to demonstrate efficacy in clinical trials. To evaluate potential interventions for intermediate AMD, we selected the sodium iodate (NaIO_3_)-induced model of retinal degeneration. This model reliably recapitulates several features of geographic atrophy, including patchy RPE loss, photoreceptor degeneration, and severe functional impairment on electroretinography (ERG), with retinal atrophy progressing over a 3-month period.

The age-related transcriptomic shifts observed in the mouse retinal pigment epithelium (RPE) by Dubey et al. provide important context for interpreting our findings [29]. Their comprehensive analysis revealed that aging induces significant changes in gene expression related to oxidative stress response, mitochondrial function, immune signaling, and lipid metabolism—all of which are pathways increasingly recognized as central to the pathophysiology of retinal degenerative conditions, including dry AMD. These data reinforce the notion that the aging RPE undergoes molecular reprogramming that may predispose it to dysfunction and disease. Our results align with this framework, further supporting the hypothesis that targeted modulation of age-sensitive pathways in the RPE may represent a viable therapeutic strategy.

Recent investigations into the aging retinal pigment epithelium (RPE) have unveiled significant epigenetic alterations that may contribute to age-related retinal diseases. Dubey et al. demonstrated a global reduction in core histones (H1, H2A, H2B, H3, and H4) and specific hypoacetylation marks (H3K14ac, H3K56ac, and H4K16ac) in aged mouse RPE/choroid tissues [30]. These changes were not observed in the neural retina, indicating a tissue-specific vulnerability of the RPE to aging. Furthermore, the study linked histone loss to the downregulation of components of the histone locus body complex, such as HINFP, and the upregulation of senescence-associated secretory phenotype (SASP) markers. These findings suggest that epigenetic dysregulation, characterized by histone deficiency and hypoacetylation, may play a pivotal role in RPE aging and the pathogenesis of conditions like age-related macular degeneration (AMD). Our study’s observations align with these findings, reinforcing the hypothesis that targeting epigenetic modifications in the RPE could be a promising therapeutic strategy for AMD.

A limitation of the NaIO_3_ model is its initial variability and the potential for partial recovery at lower doses. To ensure irreversible retinal injury and evaluate long-term therapeutic benefit, we employed a high-dose NaIO_3_ regimen (40–50 mg/kg) known to induce permanent RPE and photoreceptor damage. We also implemented multiple quantitative strategies to assess photoreceptor protection, including regional spider plots and ONL thickness averaged across the retinal span. Importantly, we evaluated therapeutic durability at a 3-month endpoint, a time point by which most photoreceptors in untreated animals had degenerated. This extended timeline more closely reflects the chronic nature of AMD than the short-term readouts commonly reported in NaIO_3_ studies.

Our data demonstrate that AAV-mediated BMI1 gene delivery via either subretinal (AAV5) or suprachoroidal (AAV8) routes significantly protects against NaIO_3_-induced retinal degeneration. Suprachoroidal delivery of AAV8.BMI1 led to greater and more consistent BMI1 protein expression compared with AAV5 subretinal delivery. Histological, imaging, and molecular analyses confirmed that BMI1 overexpression preserved photoreceptor structure (ONL thickness) and function (ERG amplitudes), establishing proof-of-concept for this approach in vivo.

The therapeutic effects of BMI1 are consistent with its known molecular functions, which include the regulation of mitochondrial bioenergetics, DNA damage repair, oxidative stress mitigation, and cellular homeostasis [5,7]. BMI1 modulates key survival pathways, such as p16INK4a/Rb and p19ARF/p53, and has previously been shown to protect cells from oxidative damage in other organ systems [6,8]. Our findings extend this protective role to the retina, with increased BMI1 levels correlating with improved photoreceptor survival and retinal function following oxidative insult [18,31].

The observed dose–response relationship, with the 1 × 10^10^ vg/eye dose providing the most robust structural and functional protection, underscores the importance of achieving sufficient transgene expression. These results are consistent with prior AAV gene therapy studies showing that higher vector doses are required to achieve therapeutic efficacy [32,33].

The suprachoroidal injection (SCI) approach employed here offers several advantages over traditional subretinal delivery, including minimally invasive administration, broader retinal coverage, and reduced procedural risk. SCI also limits systemic exposure and has been validated in both preclinical and early-phase clinical studies [34,35,36,37,38]. Our results further demonstrate that SCI enables long-term, pan-retinal expression of therapeutic transgenes, making it a compelling strategy for clinical translation [39,40]. Future pharmacokinetic studies in large animal models using clinical-grade SCI devices will help refine dose scaling and inform first-in-human trials [41].

Nonetheless, the NaIO_3_ model does not fully recapitulate all features of human AMD. It lacks hallmark features such as drusen accumulation and complement dysregulation, and the murine retina lacks a macula, limiting direct translatability to central vision outcomes. Despite these limitations, the model remains a valuable platform for evaluating gene therapies that target oxidative damage, a well-established contributor to AMD progression.

By testing two AAV serotypes, two delivery routes, and two mouse strains, we demonstrated consistent and durable retinal protection with BMI1 gene therapy. This robust body of evidence supports BMI1 as a novel target for early intervention in dry AMD.

In summary, this study established that BMI1 gene therapy, delivered via suprachoroidal injection, durably preserves retinal structure and function in a clinically relevant model of oxidative-stress-induced degeneration. Using multimodal assessment (OCT, ERG, histology, and molecular quantification), we showed that AAV8.BMI1 delivered via SCI confers broad protection, identified a therapeutic dose (1 × 10^10^ vg/eye), and validated a minimally invasive, translatable delivery method. These findings address a critical unmet need in dry AMD by combining target innovation with delivery optimization for real-world application.

## 4. Materials and Methods

### 4.1. AAV Vectors

AAV8 Stuffer was produced by Vector Builder (Chicago, IL, USA) and contained a 981 bp noncoding sequence. The BMI1 transgene in both AAV5 (SR) and AAV8 (SC) vectors is driven by the ubiquitous CAG promoter (CMV early enhancer/chicken β-actin promoter), chosen for its robust expression across retinal layers. AAV5.BMI1 was produced by Signagen (Frederick, MD, USA), and AAV8.BMI1 was produced by Forge Biologics (Grove City, OH, USA). Vectors were diluted to target concentrations using balanced salt solution (BSS; Alcon, Fort Worth, TX, USA).

### 4.2. Animals

All animals were treated in accordance with the Association for Research in Vision and Ophthalmology Statement for Use of Animals in Ophthalmic and Vision Research and in accordance with our IACUC-approved protocols. A total of 36 Balb/c mice and 20 C57BL/6 mice were used.

Balb/c mice (8 weeks old) were divided into the following groups: (1) BSS + NaIO_3_, (2) AAV5.BMI1 (1 × 10^9^ vg/eye) + NaIO_3_, and (3) untreated controls. All groups received bilateral subretinal injections followed by NaIO_3_ (50 mg/kg, IP) 4 weeks later. Animals were euthanized at 7, 21, and 105 days for tissue collection. In a second Balb/c cohort (n = 20), groups received BSS, BSS + NaIO_3_, or AAV5.BMI1 + NaIO_3_. OCT was performed on day 35, ERG was performed on day 40, and tissues were collected on day 42.

C57BL/6 mice (10 weeks old) were assigned to the following groups: (1) BSS (n = 5), (2) AAV8.stuffer (1 × 10^10^ vg/eye, n = 5), (3) AAV8.BMI1 (5 × 10^9^ vg/eye, n = 5), and (4) AAV8.BMI1 (1 × 10^10^ vg/eye, n = 5) Mice received bilateral suprachoroidal injections. NaIO_3_ (40 mg/kg, IV) was administered 4 weeks later.

### 4.3. Subretinal Injection

Mice were anesthetized using 1.5% isoflurane in oxygen. Pupils were dilated with 0.5% Cyclopentolate HCl and 10% Phenylephrine HCl, followed by topical 0.5% Proparacaine HCl. Using a 34-gauge blunt Hamilton syringe, 1 µL of vector or BSS was injected into the subretinal space under direct visualization with a surgical microscope.

### 4.4. Suprachoroidal Injection

Suprachoroidal injections were performed as described previously [12]. After pupil dilation and anesthesia, a 34-gauge needle created a partial-thickness scleral tunnel 4 mm posterior to the limbus. A 10 µL Hamilton syringe with a 34-gauge blunt-tip needle was then used to inject 1 µL of vector or BSS into the suprachoroidal space.

### 4.5. Electroretinography

Mice were dark-adapted overnight and anesthetized with ketamine (150 mg/kg) and xylazine (45 mg/kg) under dim-red light. Pupils were dilated as above. ERGs were recorded using the Celeris system (Diagnosys LLC, Lowell, MA, USA), assessing scotopic, photopic, and c-wave responses.

### 4.6. Optical Coherence Tomography

OCT imaging was conducted using the Spectralis HRA+OCT system (Heidelberg Engineering, Heidelberg, Germany) under isoflurane anesthesia. Images were processed using Eye Explorer software 1.10 with manual correction of segmentation when needed. ONL and total retinal thickness were quantified.

### 4.7. RNA Extraction and qRT-PCR

Total RNA was extracted from ocular tissue using the RNeasy Mini Kit (Qiagen, Valencia, CA, USA) and quantified using a NanoDrop spectrophotometer (Thermo Fisher Scientific, Waltham, MA, USA). cDNA was synthesized with the First Strand cDNA Synthesis Kit (Thermo Fisher Scientific), and qRT-PCR was performed using the QuantStudio 6 Pro system. Relative expression was calculated using the 2^−ΔΔCt^ method.

### 4.8. Meso Scale Discovery (MSD) Assay for BMI1 Quantification

Retina and RPE lysates were prepared in T-PER reagent (Thermo Fisher Scientific) with Halt Protease Inhibitor. Protein concentrations were measured with the BCA Protein Assay Kit. BMI1 was quantified via antigen-capture immunoassay using the Meso Scale Discovery (MSD) platform and Discovery Workbench v4.0 software.

### 4.9. Immunofluorescence and Immunohistochemistry

Eyes were fixed in 4% paraformaldehyde for 24 h at 4 °C, cryoprotected in sucrose (10%, 20%, 30%), embedded in OCT, and sectioned at 7 µm. Sections were stained with mouse anti-rhodopsin antibody (Novus Biologicals LLC, Centennial, CO, USA) and Alexa Fluor 488-conjugated goat anti-mouse IgG, with nuclei counterstained with DAPI (Invitrogen, Waltham, MA, USA). Images were captured using the EVOS M7000 system.

For IHC, sections were incubated with rabbit anti-BMI1 (Bethyl Laboratories, Montgomery, TX, USA) and goat anti-rabbit IgG (Jackson ImmunoResearch Labs, West Grove, PA, USA), then developed using the ImmPACT DAB substrate kit and counterstained with hematoxylin (Sigma-Aldrich, St. Louis, MO, USA).

### 4.10. Quantification of Outer Nuclear Layer Thickness

ONL thickness was quantified on OCT images using Eye Explorer software with manual correction. In AAV5 studies, five B-scans per eye were used to measure ONL thickness at 100 µm intervals on either side of the optic nerve. In AAV8 studies, measurements were taken at 1500 µm from the optic nerve, and spider plots were generated for topographic comparison.

### 4.11. Data Analysis

Statistical analysis was performed using GraphPad Prism 10 (GraphPad Software, Boston, MA, USA). Data are reported as mean ± SD. Comparisons between two groups were performed using the Mann–Whitney U test. For comparisons among multiple groups, one-way ANOVA with Tukey’s post hoc test was used. *p*-values < 0.05 were considered statistically significant.

## 5. Conclusions

The sodium-iodate-induced oxidative stress model provides a robust and reproducible system for evaluating retinal degeneration therapies, overcoming several limitations of transgenic models. Although acute in nature, this model effectively induces oxidative-stress-mediated RPE and photoreceptor cell death and emulates chronic oxidative stress after three months, closely mimicking key pathologic features of advanced dry AMD. Using this model, we demonstrated that suprachoroidal delivery of AAV8.BMI1 results in superior transduction efficiency and safety compared with subretinal AAV5 delivery. Across two murine species, AAV-mediated BMI1 expression provided significant protection against NaIO_3_-induced structural and functional retinal damage.

Our findings confirm that BMI1 gene therapy enhances retinal resilience by mitigating oxidative damage, preserving outer nuclear layer thickness and ERG responses. The durable expression achieved through suprachoroidal administration, combined with broad retinal distribution and minimal invasiveness, makes this a compelling approach for clinical translation.

In summary, this work established BMI1 gene therapy as a novel, mechanism-driven strategy to counteract chronic oxidative retinal degeneration. It further identified suprachoroidal AAV8 delivery as a clinically translatable platform, bridging the gap between preclinical efficacy and the urgent therapeutic need for early intervention in dry AMD with a minimally invasive route of administration.

## Figures and Tables

**Figure 1 ijms-26-05907-f001:**
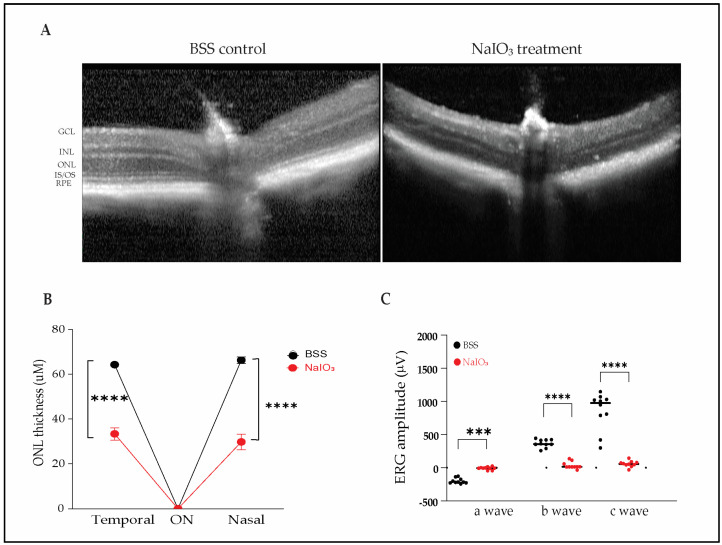
Sodium iodate induces severe retinal degeneration in mice. (**A**) Representative OCT images 4 weeks post-injection with BSS or NaIO_3_ (40 mg/kg, IV), demonstrating disruption of outer retinal layers in treated animals. GCL, ganglion cell layer; IPL, inner plexiform layer; INL, inner nuclear layer; OPL, outer plexiform layer; ONL, outer nuclear layer; ELM, external limiting membrane; IS/OS, inner/outer segments; RPE, retinal pigment epithelium. (**B**) Quantification of ONL thickness 1500 µm from the optic nerve (*n* = 10 eyes/group). (**C**) ERG recordings demonstrating complete loss of scotopic a-wave, photopic b-wave, and c-wave amplitudes in NaIO_3_-treated animals (*n* = 10 eyes/group). Data are presented as mean ± SD. *** *p* < 0.001; **** *p* < 0.0001.

**Figure 2 ijms-26-05907-f002:**
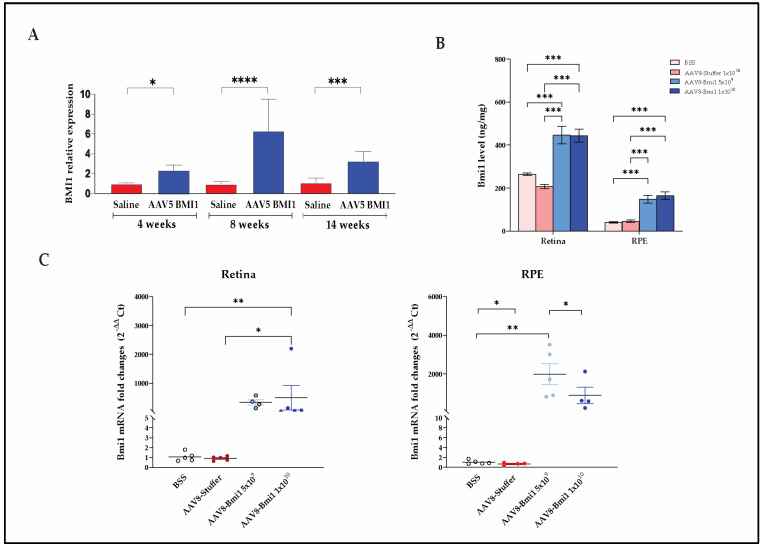
AAV-mediated BMI1 gene therapy induces durable, dose-dependent expression in murine retina and RPE. (**A**) BMI1 mRNA levels in RPE at 4, 8, and 14 weeks following subretinal AAV5.BMI1 (1 × 10^9^ vg/eye) delivery (*n* = 5 per group). (**B**) BMI1 protein levels in retina and RPE measured by MSD assay 4 weeks after suprachoroidal AAV8.BMI1 delivery at 5 × 10^9^ or 1 × 10^10^ vg/eye (*n* = 5 per group). (**C**) BMI1 mRNA expression in microdissected retina and RPE following high-dose suprachoroidal AAV8.BMI1 (1 × 10^10^ vg/eye) administration (*n* = 5 per group). Data are presented as mean ± SD. * *p* < 0.05; ** *p* < 0.01; *** *p* < 0.001; and **** *p* < 0.0001.

**Figure 3 ijms-26-05907-f003:**
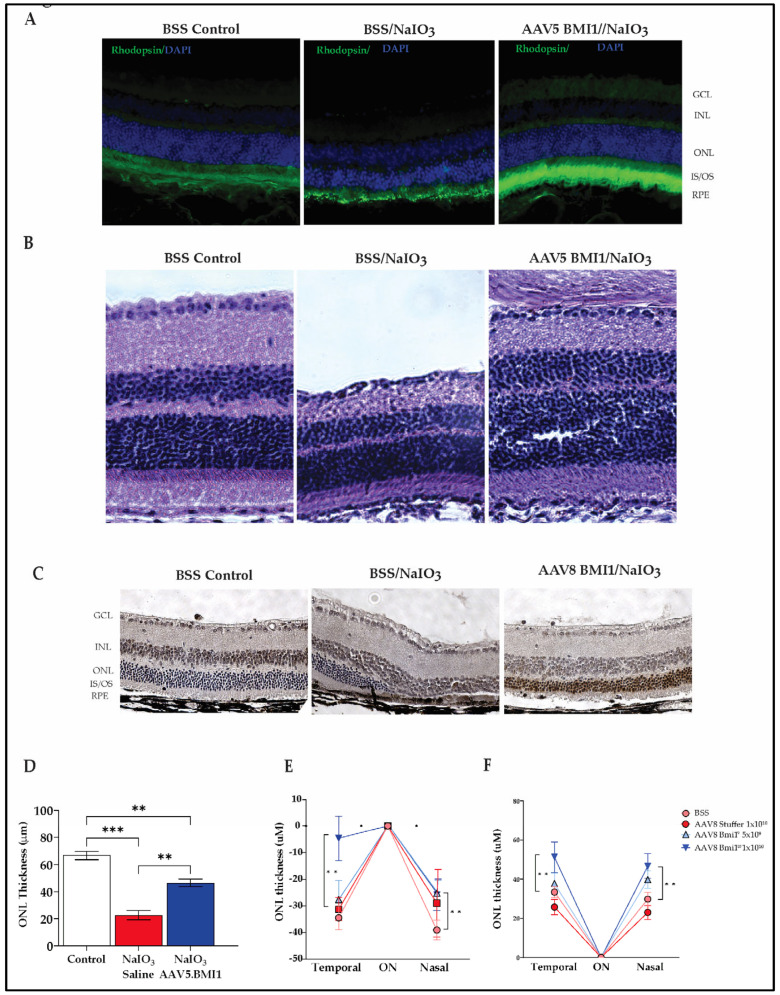
AAV-delivered *BMI1* protects against NaIO_3_-induced retinal damage. (**A**) Rhodopsin immunofluorescence (green) in retinas of BSS or AAV5.BMI1-SR-treated Balb/c mice (50 mg/kg NaIO_3_, IP; *n* = 7), showing ONL preservation in treated mice. Magnification (20×). (**B**) H&E staining of retinal cross-sections following AAV5.BMI1 or control treatment reveals preservation of retinal layers in BMI1-treated mice. Magnification (20×). (**C**) BMI1 immunohistochemical staining (brown) in C57BL/6 mice shows robust nuclear BMI1 expression in the ONL following AAV8.BMI1 treatment (40 mg/kg NaIO_3_, IV; *n* = 5). Magnification (20×). (**D**) Representative ONL measurements from H&E-stained retinas. (**E**) Quantification of ONL thickness from baseline (1500 µm from optic nerve) shows significant ONL preservation in AAV8.BMI1-treated animals. (**F**) Total ONL thickness is significantly increased in AAV8.BMI1-treated eyes compared with controls. Data are presented as mean ± SD, ** *p* < 0.01; *** *p* < 0.001; *n* = 5 per group.

**Figure 4 ijms-26-05907-f004:**
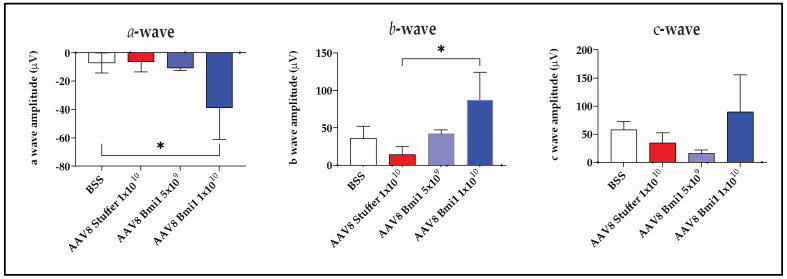
AAV8.BMI1 preserves retinal function following NaIO_3_-induced oxidative stress. Representative ERG traces recorded at baseline and 4 weeks post-NaIO_3_ injection in mice treated with BSS, AAV8.stuffer, or AAV8.BMI1 at two doses. Quantification of scotopic a-wave and photopic b-wave amplitudes demonstrates significant functional preservation in animals receiving 1 × 10^10^ vg/eye AAV8.BMI1 (*n* = 5 per group). Data are shown as mean ± SD. * *p* < 0.05.

## Data Availability

Data are contained within the article.

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
