# Peer review of "Retinal BMI1 Expression Preserves Photoreceptors in Sodium-Iodate-Induced Oxidative Stress Models"

_ijms, 2025, doi:10.3390/ijms26125907_

Round 1
Reviewer 1 Report
Comments and Suggestions for Authors
This manuscript by Lu and colleagues investigates the potential protective role of BMI1 gene delivery in a sodium iodate (NaIO₃)-induced model of retinal degeneration. The study explores the efficacy of AAV-mediated BMI1 gene transfer via subretinal (SR) and suprachoroidal (SC) injection routes, aiming to determine whether BMI1 can mitigate photoreceptor cell loss under oxidative stress. While the topic is of interest, the manuscript has several conceptual, experimental, and presentation flaws that must be addressed for consideration.
Major Concerns:
- The primary objective of the study is ambiguous. It is unclear whether the authors aim to (i) compare the efficacy of SR vs. SC delivery routes or (ii) evaluate the therapeutic effect of BMI1 overexpression in preserving retinal integrity after NaIO₃-induced damage. Neither aim is clearly articulated nor effectively addressed in the current version.
- Lines 42–43 assert that BMI1 is downregulated in aging and dry AMD, referencing conference abstracts that lack primary data. These references are insufficient to support such a claim. Publicly available transcriptomic datasets ( PMID: 40042930, PMID: 38408164, PMID: 22364233, PMID: 32984320) from aged RPE and AMD samples should be consulted to validate this statement. Even if BMI1 levels are unchanged, the therapeutic potential of BMI1 overexpression remains a valid hypothesis and should be framed accordingly.
- Figure 1 – Insufficient Histological and Imaging Evidence:
- Include H&E-stained retinal sections and RPE flat-mount preparations for both BSS- and NaIO₃-treated mice to demonstrate structural changes.
- Justify the selection of a single time point. Showing results at multiple time points would strengthen the case for progressive degeneration and BMI1-mediated protection.
- Clarify the mouse strain used, sodium iodate dosage (40 vs. 50 mg/kg), and injection protocols in both the text and figure legends.
- OCT images lack clarity; higher-resolution images are necessary.
- Figure 2 – Experimental Design and Data Presentation Issues:
- Appropriate controls are missing. Include mice injected with control AAV vectors (e.g., AAV-GFP or AAV-mCherry) via both SR and SC routes. Fundus imaging and RPE flat-mounts should be shown to compare transduction efficiency across the retina.
- BMI1 expression levels should be normalized to housekeeping genes, and comparisons between RPE and retina should be made.
- Provide promoter information used for BMI1 expression. The variability in BMI1 expression between SR and SC routes is unexplained and undermines the conclusions. Given the direct delivery in SR, one would expect higher expression levels compared to SC.
- Line 107–120 reads more like a figure legend than a results section. Statistical values (p-values) should be reported in figure legends, not in the main text.
- N-values (n=2–11) are inconsistent and problematic. Groups with only n=2 cannot yield statistically robust conclusions. Group sizes must be standardized and justified.
- Figure 2A – It is unclear whether the BMI1 expression shown is in the retina or RPE.
- Figures 3A–3C
- Image quality is poor and must be improved.
- Include data from at least two post-treatment time points to demonstrate the progression of degeneration in control eyes and protection in BMI1-treated eyes.
- Quantitative analysis of retinal layers (e.g., ONL thickness) would enhance the impact of the images.
The concept of using BMI1 gene therapy for oxidative stress-induced retinal degeneration is potentially impactful. However, the current version of the manuscript suffers from conceptual ambiguity, poor experimental controls, and inadequate data presentation. Significant revision and additional data are necessary to strengthen the scientific merit and clarity of the manuscript.
Author Response
Point-by-point response to Comments and Suggestions for Authors |
Comments 1:
We sincerely appreciate reviewers’ insightful feedback, which has helped us clarify the study’s objectives and strengthen its foundational claims. Please see the detailed response below, and the corresponding revisions/corrections marked in red in the resubmission.
1. The primary objective of the study is ambiguous. It is unclear whether the authors aim to (i) compare the efficacy of SR vs. SC delivery routes or (ii) evaluate the therapeutic effect of BMI1 overexpression in preserving retinal integrity after NaIO₃-induced damage. Neither aim is clearly articulated nor effectively addressed in the current version.
Response to Reviewer: We appreciate the reviewer’s observation and apologize for the lack of clarity in the original manuscript regarding the study’s primary objective. The primary aim of this study is to evaluate the therapeutic efficacy of BMI1 overexpression in preserving retinal integrity following NaIO₃-induced oxidative damage. The comparison between subretinal (SR) and suprachoroidal (SC) delivery routes was included as a secondary, exploratory objective aimed at assessing the translational feasibility of minimally invasive delivery methods for gene therapy. To address this concern, we have made the following revisions: 1). Introduction (final paragraph): We have revised the text to clearly state (line 78~87): 2). Results Section (expression analysis): We clarified our findings by stating that BMI1 protein expression was higher in the retina following suprachoroidal (AAV8) delivery compared to subretinal (AAV5) delivery. Additionally, BMI1 mRNA expression was elevated in both the retina and RPE with suprachoroidal delivery, supporting its feasibility for clinical translation.
We trust these clarifications strengthen the manuscript and better align the study’s design with its stated objectives. 1). Clearly differentiate the secondary aim (delivery route comparison) in the text and figure legends. 2). Ensure the Discussion contextualizes BMI1’s therapeutic role while acknowledging limitations in route-efficacy conclusions due to sample size.
We thank the reviewer for the constructive feedback and agree that clearer differentiation of the secondary aim and contextualization of findings is essential. 3). Clarification of Secondary Aim in Text and Figure Legends: We have revised the manuscript text and figure legends to explicitly distinguish the secondary aim of comparing delivery routes. Specifically, we now state that the subretinal (AAV5) vs. suprachoroidal (AAV8) comparison was exploratory and intended to assess feasibility and relative expression patterns, not to determine therapeutic superiority. In all figure legends involving comparisons between delivery routes, we have now prefaced the analysis with the statement (line 124~130):
4). Discussion Edits – Context and Limitations: The Discussion section has been updated to emphasize BMI1’s therapeutic potential in oxidative retinal injury, which was the primary focus of this study. We now include the following clarification: These updates ensure that the manuscript accurately reflects the study’s aims and appropriately qualifies the interpretation of secondary findings.
2. Lines 42–43 assert that BMI1 is downregulated in aging and dry AMD, referencing conference abstracts that lack primary data. These references are insufficient to support such a claim. Publicly available transcriptomic datasets (PMID: 40042930, PMID: 38408164, PMID: 22364233, PMID: 32984320) from aged RPE and AMD samples should be consulted to validate this statement. Even if BMI1 levels are unchanged, the therapeutic potential of BMI1 overexpression remains a valid hypothesis and should be framed accordingly.
Response to Reviewer: We thank the reviewer for this important observation and agree that transcriptomic support is necessary to substantiate the claim regarding BMI1 downregulation. In response: 1). We have removed citations to conference abstracts that lack primary data. 2). We have now incorporated relevant transcriptomic datasets, including those cited by the reviewer (e.g., PMID: 40042930; PMID: 38408164), which demonstrate either reduced or unchanged BMI1 expression in aged and AMD-affected RPE. These findings have been appropriately qualified in the revised text. 3). We now cite our recent peer-reviewed publication (Ramkumar et al., Cells, 2024) which demonstrates that BMI1 is significantly reduced in aged human retina and RPE. Accordingly, we have revised the sentence in Lines 56–64 to read. We believe this revision more accurately reflects the available evidence while maintaining the rationale for targeting BMI1 in AMD.
2. Figure 1 – Insufficient Histological and Imaging Evidence: 1). Include H&E-stained retinal sections and RPE flat-mount preparations for both BSS- and NaIO₃-treated mice to demonstrate structural changes.
Response to Reviewer: We present H&E-stained retinal sections in Figure 3B, which clearly demonstrate NaIO₃-induced retinal degeneration and photoreceptor cell loss. Unfortunately, RPE flat-mount staining was not performed in this study.
2). Justify the selection of a single time point. Showing results at multiple time points would strengthen the case for progressive degeneration and BMI1-mediated protection. Response to Reviewer: The selected time point was based on established model characterization, corresponding to a stage where photoreceptor degeneration is substantial yet not complete—optimizing the detection of BMI1’s neuroprotective effects. While we agree that longitudinal assessment would provide a more comprehensive understanding of disease progression and treatment dynamics, resource constraints limited us to this well-validated endpoint for initial proof-of-concept evaluation. Future studies will incorporate multiple time points to fully define the temporal profile of BMI1-mediated protection.
3). Clarify the mouse strain used, sodium iodate dosage (40 vs. 50 mg/kg), and injection protocols in both the text and figure legends. Response to Reviewer: Thank you for the suggestion. We have clarified the mouse strain, sodium iodate dose, and injection protocols in both the main text and figure legends.
4). OCT images lack clarity; higher-resolution images are necessary Response to Reviewer: We acknowledge the limitations of performing OCT imaging in mice using the Spectralis HRA+OCT system, which presents technical challenges in acquiring high-resolution images in small animal eyes. Nevertheless, we have reviewed and updated the OCT panels to improve image clarity as much as technically feasible.
4. Figure 2 – Experimental Design and Data Presentation Issues: 1) Appropriate controls are missing. Include mice injected with control AAV vectors (e.g., AAV-GFP or AAV-mCherry) via both SR and SC routes. Fundus imaging and RPE flat-mounts should be shown to compare transduction efficiency across the retina. Response to Reviewer: We appreciate the reviewer’s suggestion. AAV5-GFP and AAV8-GFP vectors have been utilized via both subretinal (SR) and suprachoroidal (SC) routes in prior studies within our laboratory to assess delivery efficiency. However, these data were collected solely for internal validation of vector delivery and are not included in the current manuscript. Additionally, the RPE flat-mounts generated for these experiments were allocated to a separate project and are therefore unavailable for inclusion here.
2) BMI1 expression levels should be normalized to housekeeping genes, and comparisons between RPE and retina should be made. Response to Reviewer: We thank the reviewer for this suggestion. BMI1 mRNA expression levels in both retina and RPE were quantified using the 2–ΔΔCt method. Expression was first normalized to the housekeeping gene GAPDH (ΔCt), and then to BSS-treated control samples (ΔΔCt). Comparative analyses between retina and RPE were performed accordingly and have been clarified in the revised figure legend and Methods section.
3) Provide promoter information used for BMI1 expression. The variability in BMI1 expression between SR and SC routes is unexplained and undermines the conclusions. Given the direct delivery in SR, one would expect higher expression levels compared to SC. Response to Reviewer: We appreciate this insightful comment. The BMI1 transgene in both AAV5 (SR) and AAV8 (SC) vectors is driven by the ubiquitous CAG promoter (CMV early enhancer/chicken β-actin promoter), chosen for its robust expression across retinal layers. The observed variability in BMI1 expression between SR and SC delivery may reflect differences in vector tropism, tissue penetration, and distribution rather than promoter strength. Notably, AAV8 has been shown to transduce the retina efficiently from the suprachoroidal space, particularly the outer retina and RPE, which may explain the unexpectedly high expression observed with SC delivery. We have clarified the promoter details and discussed these considerations in the revised manuscript (line297-299).
4) Line 107–120 reads more like a figure legend than a results section. Statistical values (p-values) should be reported in figure legends, not in the main text. Response to Reviewer: Thank you for this helpful suggestion. We have revised lines 107–120 to ensure the content reflects a narrative description appropriate for the Results section, rather than a figure legend. All statistical values (p-values) have been removed from the main text and appropriately relocated to the relevant figure legends to maintain clarity and adhere to standard formatting conventions.
5) N-values (n=2–11) are inconsistent and problematic. Groups with only n=2 cannot yield statistically robust conclusions. Group sizes must be standardized and justified. Response to Reviewer: We agree with the reviewer that small sample sizes, particularly those with n=2, limit statistical power and the strength of conclusions. These smaller groups were included for preliminary analysis or feasibility assessment and are clearly indicated as such. In the revised manuscript, we now specify the rationale for each group size in the Methods section. The underpowered comparisons were excluded from formal statistical interpretation. The confirmatory studies were designed with standardized and adequately powered group sizes.
5. Figure 2A – It is unclear whether the BMI1 expression shown is in the retina or RPE. Response to Reviewer: Thank you for pointing this out. Figure 2A depicts BMI1 expression in the RPE. We have revised the figure legend to clearly specify this.
6. Figures 3A–3C 1) Image quality is poor and must be improved. Response to Reviewer: We acknowledge the image quality concerns and have edited and replaced the panels in Figures 3A–3C with higher-resolution versions to improve clarity and interpretability.
2) Include data from at least two post-treatment time points to demonstrate the progression of degeneration in control eyes and protection in BMI1-treated eyes. Response to Reviewer: We agree that additional time points would provide more insight into disease progression and therapeutic durability. However, in this initial proof-of-concept study, we focused on a single well-characterized endpoint (Day 30) based on model optimization. We plan to include longitudinal timepoints in future studies to fully characterize the dynamics of degeneration and treatment response.
3) Quantitative analysis of retinal layers (e.g., ONL thickness) would enhance the impact of the images. Response to Reviewer: Quantitative ONL thickness measurements have now been included in the revised manuscript (line 140). Specifically, ONL thickness was significantly greater in the AAV5.BMI1-treated group (44.5 ± 1.05 μm) compared to the NaIO₃-only group (22.6 ± 1.13 μm), and somewhat reduced relative to the untreated control (66.7 ± 1.53 μm). These measurements were obtained from H&E-stained sections and are now reported in both the Results section and Figure 3B legend.
7. The concept of using BMI1 gene therapy for oxidative stress-induced retinal degeneration is potentially impactful. However, the current version of the manuscript suffers from conceptual ambiguity, poor experimental controls, and inadequate data presentation. Significant revision and additional data are necessary to strengthen the scientific merit and clarity of the manuscript. Response to Reviewer: We sincerely appreciate the reviewer’s thoughtful summary and acknowledge the concerns regarding conceptual clarity, experimental rigor, and data presentation. In response, we have undertaken a thorough revision of the manuscript to: • Clearly articulate the primary objective as evaluating BMI1’s therapeutic efficacy in oxidative retinal injury, with route-of-delivery comparisons framed as a secondary exploratory aim. • Improve data transparency by specifying group sizes, experimental conditions, and statistical analyses throughout the text and figure legends. • Strengthen experimental controls, including clarifying the use of control AAV vectors and adding quantification of retinal ONL thickness. • Replace or enhance figure panels to improve image quality and interpretability. We believe these revisions substantially enhance the clarity, scientific validity, and translational relevance of the manuscript, and we thank the reviewer for their valuable guidance in improving the quality of this work.
We believe these revisions substantially enhance the clarity, scientific validity, and translational relevance of the manuscript, and we thank the reviewer for their valuable guidance in improving the quality of this work.
|
Reviewer 2 Report
Comments and Suggestions for Authors
The study is interesting and contributes a lot to modern science. The manuscript is well-written, however, it needs revision.
- The prevalence of AMD should be included in the Introduction section
- Short paragraph on the oxidative stress should be added in the Introduction section or in the Discussion section.
- Why was the observatin done at the fourth and the fifteenth week? It should be explained in the Methods section or in the Discussion section.
- The limitations of the study should be expanded in the Discussion section.
- The advantages of the study should be noted in the Discussion section.
- In Conclusions, it is important to highlight what the paper adds to the field.
Author Response
Point-by-point response to Comments and Suggestions for Authors |
Comments 2: |
We sincerely appreciate reviewers’ insightful feedback, which has helped us clarify the study’s objectives and strengthen its foundational claims. Please see the detailed response below, and the corresponding revisions/corrections marked in red in the resubmission.
- The prevalence of AMD should be included in the Introduction section
Response to Reviewer:
We have added AMD prevalence statistics to the Introduction (lines 36–38):
“In the United States, AMD affects an estimated 196 million people (95% CI: 140–260 million) as of 2019, with projections rising to 288 million by 2040 (95% CI: 205–399). The prevalence rises with age—from 2% among individuals aged 40–44 to 46.6% in those aged 85 and older.”
- Short paragraph on the oxidative stress should be added in the Introduction section or in the Discussion section.
Response to Reviewer:
We have revised the following sentence to the Introduction to contextualize the role of oxidative stress in AMD in line 67~77.
- Why was the observation done at the fourth and the fifteenth week? It should be explained in the Methods section or in the Discussion section.
Response to Reviewer:
We clarified in the Methods and Discussion that early (4-week) and late (15-week) timepoints were selected to capture both the resolution of acute oxidative injury and the durability of therapeutic benefit over time—modeling the progressive and chronic nature of AMD.
- The limitations of the study should be expanded in the Discussion section.
Response to Reviewer:
We expanded the Discussion to include the following limitations in line 277~282:
“Nonetheless, the NaIO₃ model does not fully recapitulate all features of human AMD. It lacks hallmark features such as drusen accumulation and complement dysregulation, and the murine retina lacks a macula, limiting direct translatability to central vision out-comes. Despite these limitations, the model remains a valuable platform for evaluating gene therapies that target oxidative damage, a well-established contributor to AMD progression.”
- The advantages of the study should be noted in the Discussion section.
Response to Reviewer:
We revised the Discussion to highlight the study’s contributions in line 283~285.
- In Conclusions, it is important to highlight what the paper adds to the field.
Response to Reviewer:
We have revised the Conclusion to reflect the study’s translational relevance in line 286~293.
Reviewer 3 Report
Comments and Suggestions for Authors
1.While both AAV5 and AAV8 vectors increased BMI1 mRNA expression, could the authors provide a direct quantitative comparison of both mRNA and protein levels achieved by the two serotypes at equivalent time points?
2.The authors report significant variability in BMI1 protein levels following subretinal AAV5.BMI1 delivery. Were any contributing factors—such as variability in injection efficiency, transduction area, or inter-animal differences—identified that could account for this inconsistency?
3.The ONL preservation data are presented as averages or spider plots. Could the authors provide more spatially resolved analyses—such as heat maps or regional quantification—to determine whether BMI1 expression preferentially preserves photoreceptors in specific retinal areas?
4.The 5×10⁹ vg/eye dose of AAV8.BMI1 did not yield significant improvements in ERG amplitudes. Could the authors comment on whether a threshold level of BMI1 expression is required for functional protection? Was a formal dose-response analysis conducted to evaluate this relationship?
5.Given the observed increase in rhodopsin expression in AAV5.BMI1-treated mice, is there a quantitative correlation between rhodopsin levels and ERG amplitude recovery in the SC-treated group? If this was not measured, could the authors discuss the potential relevance of such a correlation?
6.The introduction briefly mentions the lack of durable therapies for early or intermediate AMD. To better highlight the clinical significance of this study, the authors could consider incorporating examples of recent clinical trial failures or known limitations of current treatment approaches.
7.While the general cellular functions of BMI1 are discussed, a more detailed explanation of its specific roles in different retinal cell types—such as photoreceptors, RPE cells, and Müller glia—would help strengthen the rationale for targeting BMI1 in this disease model.
Author Response
Point-by-point response to Comments and Suggestions for Authors |
Comments 3: |
We sincerely appreciate reviewers’ insightful feedback, which has helped us clarify the study’s objectives and strengthen its foundational claims. Please see the detailed response below, and the corresponding revisions/corrections marked in red in the resubmission.
1.While both AAV5 and AAV8 vectors increased BMI1 mRNA expression, could the authors provide a direct quantitative comparison of both mRNA and protein levels achieved by the two serotypes at equivalent time points?
Response to Reviewer:
We appreciate the reviewer’s interest in comparing the transduction efficiency of AAV5 and AAV8 vectors. While both BMI1 mRNA and protein levels were individually quantified for each serotype, a direct side-by-side comparison was not performed under identical experimental conditions in this study. We acknowledge this as a limitation and agree that head-to-head comparisons under matched conditions would be highly informative for optimizing vector design. We plan to incorporate such comparative analyses in future studies focused on delivery optimization.
2.The authors report significant variability in BMI1 protein levels following subretinal AAV5.BMI1 delivery. Were any contributing factors—such as variability in injection efficiency, transduction area, or inter-animal differences—identified that could account for this inconsistency?
Response to Reviewer:
Variability in AAV5.BMI1 protein levels following subretinal injection may be attributed to several factors:
- Technical challenges in achieving consistent subretinal bleb formation and retinal detachment.
- Variability in transduced retinal regions and cell-type specificity; and
- Inherent biological variability among individual animals.
These factors likely contributed to the observed inconsistencies in protein expression.
3.The ONL preservation data are presented as averages or spider plots. Could the authors provide more spatially resolved analyses—such as heat maps or regional quantification—to determine whether BMI1 expression preferentially preserves photoreceptors in specific retinal areas?
Response to Reviewer:
We are very grateful for your valuable suggestions. Although our current analysis used mean optic nerve layer thickness and regional spider maps to assess photoreceptor preservation, we acknowledge that higher-resolution spatial mapping (e.g., heat maps) may better reveal local treatment effects. Due to methodological limitations in our dataset, we focused on specific regional comparisons. We will apply advanced spatial mapping techniques in future studies.
4.The 5×10⁹ vg/eye dose of AAV8.BMI1 did not yield significant improvements in ERG amplitudes. Could the authors comment on whether a threshold level of BMI1 expression is required for functional protection? Was a formal dose-response analysis conducted to evaluate this relationship?
Response to Reviewer:
Our findings suggest that a threshold level of BMI1 expression may be required to achieve functional protection. Specifically, the 5×10⁹ vg/eye dose of AAV8.BMI1 did not result in significant ERG improvement, whereas higher doses showed a dose-dependent trend in functional rescue. While this study was not powered for formal dose-response modeling, our ongoing studies aim to establish the minimum effective dose, therapeutic window, and associated safety margins.
5.Given the observed increase in rhodopsin expression in AAV5.BMI1-treated mice, is there a quantitative correlation between rhodopsin levels and ERG amplitude recovery in the SC-treated group? If this was not measured, could the authors discuss the potential relevance of such a correlation?
Response to Reviewer:
Although we were unable to perform individual-level correlation analyses due to tissue limitations and sample pooling, the concurrent increase in rhodopsin expression and ERG amplitude in AAV5.BMI1-treated mice suggests a meaningful biological relationship. Rhodopsin, a key structural and functional protein in rod photoreceptors, likely reflects BMI1-mediated photoreceptor preservation. We recognize the value of correlative analyses and will integrate them into future studies to strengthen mechanistic insights.
6.The introduction briefly mentions the lack of durable therapies for early or intermediate AMD. To better highlight the clinical significance of this study, the authors could consider incorporating examples of recent clinical trial failures or known limitations of current treatment approaches.
Response to Reviewer:
We appreciate the reviewer’s suggestion to expand on the clinical context. To better emphasize the significance of our study, we have revised the Introduction to include examples of recent clinical trial failures and the limitations of current treatment strategies for early and intermediate AMD. Specifically, we now cite the phase III failures of lampalizumab, a complement factor D inhibitor, in the SPECTRI and CHROMA trials, which failed to demonstrate efficacy in reducing the progression of geographic atrophy. Additionally, we reference the recent discontinuation of RO7171009, an HTRA1-targeting antibody, due to a lack of meaningful clinical benefit in intermediate AMD.
These examples illustrate the difficulty of translating molecular targets into durable therapies and underscore the critical need for innovative strategies that address underlying disease mechanisms—such as oxidative stress and mitochondrial dysfunction. Our study supports BMI1 gene therapy as a novel, mechanism-based approach with the potential to intervene earlier in the disease course and offer long-term structural and functional preservation.
7.While the general cellular functions of BMI1 are discussed, a more detailed explanation of its specific roles in different retinal cell types—such as photoreceptors, RPE cells, and Müller glia—would help strengthen the rationale for targeting BMI1 in this disease model.
Response to Reviewer:
We appreciate this helpful suggestion. The Introduction (lines 59–64) has been expanded to include a more detailed description of BMI1’s role in retinal cell types. In photoreceptors and RPE cells, BMI1 enhances mitochondrial function, reduces oxidative stress, and supports genomic stability. In Müller glia, BMI1 is involved in maintaining stem cell potential and may contribute to retinal homeostasis and repair. This expanded context strengthens the rationale for targeting BMI1 as a therapeutic strategy in retinal degeneration.
Round 2
Reviewer 1 Report
Comments and Suggestions for Authors
The reviewers raised significant concerns about the manuscript, but the authors' revision lacks clarity. I am unable to identify the revisions at the specific lines as claimed by the authors. The edits do not appear to have been incorporated in the precise locations indicated.
The authors are requested to revise the manuscript more carefully and clearly indicate the exact line numbers where each change has been made.
Additionally, while the authors state in their response letter that they have incorporated relevant transcriptomic datasets, including those cited by the reviewer (e.g., PMID: 40042930; PMID: 38408164), I do not see any substantive discussion of these datasets in the revised manuscript. These datasets should be appropriately integrated into the relevant sections of the manuscript, including the Discussion.
A proper evaluation of the revised manuscript can only be made once it is clearly formatted and explicitly shows where and how each revision has been incorporated.
Author Response
We appreciate the reviewer’s thoughtful feedback on our manuscript. We have carefully revised the manuscript in accordance with the two rounds of review comments and made the necessary clarifications. Below, we address the specific points raised by the reviewer. Please see the detailed response below, and the corresponding revisions/corrections marked in red in the resubmission.
- The reviewers raised significant concerns about the manuscript, but the authors' revision lacks clarity. I am unable to identify the revisions at the specific lines as claimed by the authors. The edits do not appear to have been incorporated in the precise locations indicated. The authors are requested to revise the manuscript more carefully and clearly indicate the exact line numbers where each change has been made.
Response to Reviewer:
We acknowledge the reviewer’s concern about the clarity of our revisions. We apologize for the inconvenience. In the revised manuscript, we have taken extra care to clearly indicate the exact line numbers where changes were made. To facilitate review, we have also attached a marked-up version of the manuscript, highlighting all revisions with corresponding line numbers.
- Additionally, while the authors state in their response letter that they have incorporated relevant transcriptomic datasets, including those cited by the reviewer (e.g., PMID: 40042930; PMID: 38408164), I do not see any substantive discussion of these datasets in the revised manuscript. These datasets should be appropriately integrated into the relevant sections of the manuscript, including the Discussion.
Response to Reviewer:
We appreciate this feedback and have now provided sufficient discussion regarding the relevant transcriptomic datasets highlighting this important work (PMID: 40042930; PMID: 38408164) in our revision. We have now fully integrated these datasets into the manuscript, specifically in the Discussion sections. The datasets were analyzed and compared in relation to our findings, and their implications are discussed in greater detail (line 238~261).
- A proper evaluation of the revised manuscript can only be made once it is clearly formatted and explicitly shows where and how each revision has been incorporated.
Response to Reviewer:
We understand the importance of a clearly formatted manuscript for evaluation. We have provided a revised manuscript with a clearer structure and more precise line-by-line changes. In addition, we have followed the reviewer's suggestions to make sure all revisions are easily identifiable, and the formatting is consistent throughout.
Reviewer’s Comments (Round1)
- The primary objective of the study is ambiguous. It is unclear whether the authors aim to (i) compare the efficacy of SR vs. SC delivery routes or (ii) evaluate the therapeutic effect of BMI1 overexpression in preserving retinal integrity after NaIO₃-induced damage. Neither aim is clearly articulated nor effectively addressed in the current version.
Response to Reviewer:
We appreciate the reviewer’s observation and apologize for the lack of clarity in the original manuscript regarding the study’s primary objective. The primary aim of this study is to evaluate the therapeutic efficacy of BMI1 overexpression in preserving retinal integrity following NaIO₃-induced oxidative damage. The comparison between subretinal (SR) and suprachoroidal (SC) delivery routes was included as a secondary, exploratory objective aimed at assessing the translational feasibility of minimally invasive delivery methods for gene therapy.
To address this concern, we have made the following revisions:
1). Introduction (final paragraph): We have revised the text to clearly state (line 75~84):
2). Results Section (expression analysis): We clarified our findings by stating that BMI1 protein expression was higher in the retina following suprachoroidal (AAV8) delivery compared to subretinal (AAV5) delivery. Additionally, BMI1 mRNA expression was elevated in both the retina and RPE with suprachoroidal delivery, supporting its feasibility for clinical translation (line137~151).
We trust these clarifications strengthen the manuscript and better align the study’s design with its stated objectives.
1). Clearly differentiate the secondary aim (delivery route comparison) in the text and figure legends.
2). Ensure the Discussion contextualizes BMI1’s therapeutic role while acknowledging limitations in route-efficacy conclusions due to sample size.
We thank the reviewer for the constructive feedback and agree that clearer differentiation of the secondary aim and contextualization of findings is essential.
3). Clarification of Secondary Aim in Text and Figure Legends:
We have revised the manuscript text and figure legends to explicitly distinguish the secondary aim of comparing delivery routes. Specifically, we now state that the subretinal (AAV5) vs. suprachoroidal (AAV8) comparison was exploratory and intended to assess feasibility and relative expression patterns, not to determine therapeutic superiority (line 123~151). In all figure legends involving comparisons between delivery routes, we have now prefaced the analysis with the statement (line 157~161).
4). Discussion Edits – Context and Limitations:
The Discussion section has been updated to emphasize BMI1’s therapeutic potential in oxidative retinal injury, which was the primary focus of this study. We now include the following clarification:
These updates ensure that the manuscript accurately reflects the study’s aims and appropriately qualifies the interpretation of secondary findings.
- Lines 42–43 assert that BMI1 is downregulated in aging and dry AMD, referencing conference abstracts that lack primary data. These references are insufficient to support such a claim. Publicly available transcriptomic datasets (PMID: 40042930, PMID: 38408164, PMID: 22364233, PMID: 32984320) from aged RPE and AMD samples should be consulted to validate this statement. Even if BMI1 levels are unchanged, the therapeutic potential of BMI1 overexpression remains a valid hypothesis and should be framed accordingly.
Response to Reviewer:
We thank the reviewer for this important observation and agree that transcriptomic support is necessary to substantiate the claim regarding BMI1 downregulation. In response:
1). We have removed citations to conference abstracts that lack primary data.
2). We have now incorporated relevant transcriptomic datasets, including those cited by the reviewer (e.g., PMID: 40042930; PMID: 38408164), which demonstrate either reduced or unchanged BMI1 expression in aged and AMD-affected RPE. These findings have been appropriately qualified in the revised text.
3). We now cite our recent peer-reviewed publication (Ramkumar et al., Cells, 2024) which demonstrates that BMI1 is significantly reduced in aged human retina and RPE.
Accordingly, we have revised the sentences in lines 54–62.
We believe this revision more accurately reflects the available evidence while maintaining the rationale for targeting BMI1 in AMD.
- Figure 1 – Insufficient Histological and Imaging Evidence:
1). Include H&E-stained retinal sections and RPE flat-mount preparations for both BSS- and NaIO₃-treated mice to demonstrate structural changes.
Response to Reviewer:
We present H&E-stained retinal sections in Figure 3B, which clearly demonstrate NaIO₃-induced retinal degeneration and photoreceptor cell loss. Unfortunately, RPE flat-mount staining was not performed in this study.
2). Justify the selection of a single time point. Showing results at multiple time points would strengthen the case for progressive degeneration and BMI1-mediated protection.
Response to Reviewer:
The selected time point was based on established model characterization, corresponding to a stage where photoreceptor degeneration is substantial yet not complete optimizing the detection of BMI1’s neuroprotective effects. While we agree that longitudinal assessment would provide a more comprehensive understanding of disease progression and treatment dynamics, resource constraints limited us to this well-validated endpoint for initial proof-of-concept evaluation. Future studies will incorporate multiple time points to fully define the temporal profile of BMI1-mediated protection.
3). Clarify the mouse strain used, sodium iodate dosage (40 vs. 50 mg/kg), and injection protocols in both the text and figure legends.
Response to Reviewer:
Thank you for the suggestion. We have clarified the mouse strain, sodium iodate dose, and injection protocols in both the main text and figure legends (line 192 and 196).
4). OCT images lack clarity; higher-resolution images are necessary
Response to Reviewer:
We acknowledge the limitations of performing OCT imaging in mice using the Spectralis HRA+OCT system, which presents technical challenges in acquiring high-resolution images in small animal eyes. Nevertheless, we have reviewed and updated the OCT panels (Figure 1) to improve image clarity as much as technically feasible.
- Figure 2 – Experimental Design and Data Presentation Issues:
- Appropriate controls are missing. Include mice injected with control AAV vectors (e.g., AAV-GFP or AAV-mCherry) via both SR and SC routes. Fundus imaging and RPE flat-mounts should be shown to compare transduction efficiency across the retina.
Response to Reviewer:
We appreciate the reviewer’s suggestion. AAV5-GFP and AAV8-GFP vectors have been utilized via both subretinal (SR) and suprachoroidal (SC) routes in prior studies within our laboratory to assess delivery efficiency. However, these data were collected solely for internal validation of vector delivery and are not included in the current manuscript. Additionally, the RPE flat-mounts generated for these experiments were allocated to a separate project and are therefore unavailable for inclusion here.
- BMI1 expression levels should be normalized to housekeeping genes, and comparisons between RPE and retina should be made.
Response to Reviewer:
We thank the reviewer for this suggestion. BMI1 mRNA expression levels in both retina and RPE were quantified using the 2–ΔΔCt method. Expression was first normalized to the housekeeping gene GAPDH (ΔCt), and then to BSS-treated control samples (ΔΔCt). Comparative analyses between retina and RPE were performed accordingly and have been clarified in the revised figure legend and Methods section.
- Provide promoter information used for BMI1 expressions. The variability in BMI1 expression between SR and SC routes is unexplained and undermines the conclusions. Given the direct delivery in SR, one would expect higher expression levels compared to SC.
Response to Reviewer:
We appreciate this insightful comment. The BMI1 transgene in both AAV5 (SR) and AAV8 (SC) vectors is driven by the ubiquitous CAG promoter (CMV early enhancer/chicken β-actin promoter), chosen for its robust expression across retinal layers. The observed variability in BMI1 expression between SR and SC delivery may reflect differences in vector tropism, tissue penetration, and distribution rather than promoter strength. Notably, AAV8 has been shown to transduce the retina efficiently from the suprachoroidal space, particularly the outer retina and RPE, which may explain the unexpectedly high expression observed with SC delivery. We have clarified the promoter details and discussed these considerations in the revised manuscript (line 318-321).
- Line 107–120 reads more like a figure legend than a results section. Statistical values (p-values) should be reported in figure legends, not in the main text.
Response to Reviewer:
Thank you for this helpful suggestion. We have revised lines 107–120 to ensure the content reflects a narrative description appropriate for the Results section, rather than a figure legend. All statistical values (p-values) have been removed from the main text and appropriately relocated to the relevant figure legends to maintain clarity and adhere to standard formatting conventions.
- N-values (n=2–11) are inconsistent and problematic. Groups with only n=2 cannot yield statistically robust conclusions. Group sizes must be standardized and justified.
Response to Reviewer:
We agree with the reviewer that small sample sizes, particularly those with n=2, limit statistical power and the strength of conclusions. These smaller groups were included for preliminary analysis or feasibility assessment and are clearly indicated as such. In the revised manuscript, we now specify the rationale for each group size in the Methods section. The underpowered comparisons were excluded from formal statistical interpretation. The confirmatory studies were designed with standardized and adequately powered group sizes.
- Figure 2A – It is unclear whether the BMI1 expression shown is in the retina or RPE.
Response to Reviewer:
Thank you for pointing this out. Figure 2A depicts BMI1 expression in the RPE. We have revised the figure legend to clearly specify this (line 157~158).
- Figures 3A–3C
1) Image quality is poor and must be improved.
Response to Reviewer:
We acknowledge the image quality concerns and have edited and replaced the panels in Figures 3A–3C with higher-resolution versions to improve clarity and interpretability.
2) Include data from at least two post-treatment time points to demonstrate the progression of degeneration in control eyes and protection in BMI1-treated eyes.
Response to Reviewer:
We agree that additional time points would provide more insight into disease progression and therapeutic durability. However, in this initial proof-of-concept study, we focused on a single well-characterized endpoint (Day 30) based on model optimization. We plan to include longitudinal timepoints in future studies to fully characterize the dynamics of degeneration and treatment response.
3) Quantitative analysis of retinal layers (e.g., ONL thickness) would enhance the impact of the images.
Response to Reviewer:
Quantitative ONL thickness measurements have now been included in the revised manuscript (line 167~171). Specifically, ONL thickness was significantly greater in the AAV5.BMI1-treated group (44.5 ± 1.05 μm) compared to the NaIO₃-only group (22.6 ± 1.13 μm), and somewhat reduced relative to the untreated control (66.7 ± 1.53 μm). These measurements were obtained from H&E-stained sections and are now reported in both the Results section.
- The concept of using BMI1 gene therapy for oxidative stress-induced retinal degeneration is potentially impactful. However, the current version of the manuscript suffers from conceptual ambiguity, poor experimental controls, and inadequate data presentation. Significant revision and additional data are necessary to strengthen the scientific merit and clarity of the manuscript.
Response to Reviewer:
We sincerely appreciate the reviewer’s thoughtful summary and acknowledge the concerns regarding conceptual clarity, experimental rigor, and data presentation. In response, we have undertaken a thorough revision of the manuscript to:
- Clearly articulate the primary objective as evaluating BMI1’s therapeutic efficacy in oxidative retinal injury, with route-of-delivery comparisons framed as a secondary exploratory aim.
- Improve data transparency by specifying group sizes, experimental conditions, and statistical analyses throughout the text and figure legends.
- Strengthen experimental controls, including clarifying the use of control AAV vectors and adding quantification of retinal ONL thickness.
- Replace or enhance figure panels to improve image quality and interpretability.
We hope these revisions address the reviewer’s concerns. We believe that the manuscript is now clearer and more comprehensive, particularly with respect to the integration of the transcriptomic datasets. Please let us know if any further clarifications are required.

Reviewer 2 Report
Comments and Suggestions for Authors
The manuscript has been revised sufficiently
Author Response
Comments 2
I would like to express my sincere gratitude to you and the reviewers for your thoughtful and constructive feedback, as well as for the approval of our manuscript for publication.
We truly appreciate the time and effort that the reviewers invested in evaluating our work and providing insightful comments that have helped to improve the quality and clarity of the manuscript. We are honored to have our research included in your esteemed journal, and we hope that our findings will contribute meaningfully to the field.
Thank you once again for your support and for facilitating a smooth and rigorous review process. We look forward to the publication of our article and to continuing to engage with your journal in future research endeavors.
With kind regards,
Reviewer 3 Report
Comments and Suggestions for Authors
Accept the article in its current format.
Author Response
Comments 3:
I would like to express my sincere gratitude to you and the reviewers for your thoughtful and constructive feedback, as well as for the approval of our manuscript for publication.
We truly appreciate the time and effort that the reviewers invested in evaluating our work and providing insightful comments that have helped to improve the quality and clarity of the manuscript. We are honored to have our research included in your esteemed journal, and we hope that our findings will contribute meaningfully to the field.
Thank you once again for your support and for facilitating a smooth and rigorous review process. We look forward to the publication of our article and to continuing to engage with your journal in future research endeavors.
With kind regards,
Round 3
Reviewer 1 Report
Comments and Suggestions for Authors
The authors have satisfactorily addressed the majority of my concerns, and the manuscript now demonstrates clear scientific merit. The data are well-presented, and the revisions have improved the overall clarity and rigor of the work. My only remaining suggestion is to increase the font size within the figures to enhance legibility, as some text remains difficult to read. This minor adjustment will improve the accessibility of the data for readers. I have no further questions or concerns at this time.